# Physicochemical Properties and Fine Structure of Starch in Jinong Xiangruan 1 and DGR1 Soft Rice Varieties Cultivated in Different Regions of China

Zhuoyi Hua [1,†], Zubair Iqbal [2,†], Yu Han [1], Chenyang Wu [1], Zhongyou Pei [1], Xin Zhang [1], Jing Sun [3], Mingnan Qu [2,*] and Zhibin Li [1,*]

[1] Department of Agriculture and Resources and Environment, Tianjin Agricultural University, Tianjin 300384, China; huazhuoyi1288@163.com (Z.H.); hy18812778669@163.com (Y.H.); 15681916112@163.com (C.W.); zhongyoupei@tjau.edu.cn (Z.P.); tnzhangxin@163.com (X.Z.)

[2] Jiangsu Key Laboratory of Crop Cultivation and Physiology, Agricultural College of Yangzhou University, Yangzhou 225009, China; 008615@yzu.edu.cn

[3] Enterprises Key Laboratory of Hybrid Japonica Rice, Tianjin 300457, China; stept2009@163.com

* Correspondence: qmn@yzu.edu.cn (M.Q.); lizhibinwork@163.com (Z.L.)

[†] These authors contributed equally to this work.

**Abstract:** Rice, a staple food for billions around the globe, is cultivated in numerous forms. Among them, soft rice is well known, which is characterized by its tender, creamy consistency and desirable texture. In this study, we examined the physicochemical properties and fine structure of starch from two soft rice varieties, Jinong Xiangruan 1 and DGR1, cultivated in different regions in China (Baodi District, Tianjin City; Liaoning Province; and Fengyang City, Anhui Province). The aim was to understand how amylopectin content (AC) influences rice quality. This research aims to bridge the knowledge gap regarding the role of amylopectin in determining rice's adhesive consistency and viscosity. Significant regional differences were observed in yield components such as the number of grains per panicle, seed setting rates and 1000-grain weight, with Liaoning generally showing higher performance metrics compared to other regions. Physicochemical analysis highlighted that though glue consistency and taste values showed little regional variation, AC significantly influenced rice hardness and viscosity. Rapid Visco Analyzer (RVA) profile analysis further demonstrated distinct differences in viscosity characteristics, underscoring the regional impacts on starch behavior. Additionally, molecular weight distribution and amylopectin chain length analysis, conducted via SEC-MALLS-RI and ICS ion chromatography, revealed notable differences in starch composition across varieties and locations. The findings suggest that environmental conditions play a crucial role in defining starch characteristics and, consequently, the eating quality of rice. This provides valuable insights for breeding high-quality japonica rice with broad adaptability.

**Keywords:** soft rice; amylose content; food quality; starch structure





## 1. Introduction

Starch is the predominant component of rice, primarily comprising the glucose polymers amylose and amylopectin. Characterized by its highly branched structure, amylopectin has short α-1, 4-linked glucosyl chains connected by α-1,6 bonds, which significantly influence the texture of rice post-cooking [1]. Research indicates that while amylose content (AC) may be similar across some rice varieties, variations in taste often arise from differences in amylopectin's fine structure. These include factors like chain length distribution and molecular weight [2]. Furthermore, numerous studies suggest that these structural details play a key role in starch's functional characteristics, impacting attributes such as gelatinization and viscosity [3].

The distribution of amylopectin chain length in rice defines its basic structure, which includes three types of chains: A, B and C. Chain A, the outermost chain, features a

polymerization degree (DP) of 6–12 and is connected to the B chain via an α-1, 6-glucoside bond [4]. This chain does not form a branched structure. The B chain is categorized by average chain length into three subtypes: B1 (DP13-24), B2 (DP25-36) and B3 (DP > 36). Chains A and B1 comprise a short-chain structure, known as the S chain, while B2 and B3 form the long chain structure [5]. In both low-amylose and waxy rice varieties, the molecular ratio between A and B chains is approx. 1.1:1.5. Current research on the fine structure of rice amylopectin and its effect on rice quality traits is limited, suggesting potential areas for improvement [6,7]. Studies have primarily focused on the distribution of amylopectin chain lengths, observing how variations in polymerization degree of short, medium and long chains affect starch's physicochemical properties.

Previous research indicates that the gelatinization characteristics of starch are closely linked to distribution of amylopectin chain lengths, although they are not significantly associated with adhesive consistency or the RVA eigenvalue index. More extended chains in amylopectin tend to harden the rice texture [8,9]. When amylopectin has a high proportion of long chains (DP37-63), the starch exhibits lower peak viscosity and higher setback viscosity (SBV), resulting in poorer rice taste and harder texture. The intertwining and binding of amylopectin on the starch surface enhance its strength and elasticity, leading to a non-sticky and chewy rice texture [10].

Southern China has gradually become a key area for further expanding high-quality rice production due to its warm and humid climate and excellent conditions of sufficient temperature and light, while the northern regions primarily cultivate japonica rice, especially Heilongjiang, Jilin and Liaoning provinces in northeast China. In recent years, global warming has caused extreme weather events to occur from time to time, including damage from both low and high temperature, which have affected rice production. External factors such as the geographical environment of rice growth and cultivation conditions also impact the starch structure and physicochemical properties of rice. These conditions have a certain impact on the quality of rice grain starch at maturity, contributing to the gap in rice quality between southern and northern regions. In 2022, the average daily maximum surface temperature in Anhui Province, China, reached about 40 °C. It adversely affected the growth and development of rice, especially during the rice filling and maturation stage, leading to overall low quality of soft rice production.

Currently, research on the cultivation of high-quality rice varieties suitable for both northern and southern China is insufficient. Despite ongoing research, the role of amylopectin's fine structure in rice quality remains underexplored, particularly regarding how it influences physicochemical traits like adhesive consistency and viscosity. This gap highlights the need for further study on high-quality rice varieties that perform consistently across various climatic zones in China. Therefore, this study focuses on two soft rice varieties, Jinong Xiangruan 1 and DGR1, which differ in amylopectin content (2–5% and 6–12%, respectively). By comparing these varieties cultivated in Baodi District, Tianjin City; Liaoning Province; and Fengyang City, Anhui Province, we aim to elucidate the influence of environmental conditions on their physicochemical properties and taste profiles. The insights gained will guide the breeding of new japonica rice varieties with broad adaptability and superior quality.

## 2. Materials and Methods

### 2.1. Experimental Location and Materials

The experiment utilized two soft rice varieties, Jinong Xiangruan 1 and DGR1, with significant differences in AC. These varieties were used as experimental materials. Planting was conducted in three different ecological regions during the normal season of 2022: Baodi District of Tianjin City, Zhuanghe City of Liaoning Province and Fengyang City of Anhui Province. Fengyang County, Chuzhou City, Anhui Province, is located in the northeast of Anhui Province, China, the south bank of the middle reaches of the Huaihe River, north latitude 32°37′–33°03′, east longitude 117°19′–117°57′. The average frost-free period over the years is 212 days; Baodi District of Tianjin is located in the north of

Tianjin, China, with geographical coordinates of east longitude 117°8′~117°40′ and north latitude 39°21′~39°50′. The average frost-free period in the past years is about 184 days; Zhuanghe County, Dalian City, Liaoning Province, China, is located in the southeast of Liaodong Peninsula, with geographical coordinates of east longitude 122°29~123°31 and north latitude 39°25′~40°12′. The average frost-free period is 165 days. The experiment employed a mechanical transplantation method. Each plot length was 5 m with 10 rows spaced 30 cm apart and a plant spacing of 16 cm, transplanting 3–5 seedlings per hole. Irrigation and fertilizer management were scheduled following local farming practices. Pest and disease control measures were applied as required.

*2.2. Raw Material Collection*

This experiment compared the differences in taste quality and physicochemical indicators of two soft rice varieties under different climatic conditions such as photoperiod and temperature. It aimed to further study the fine structure of soft rice starch and identify key factors affecting taste quality. The nursery was sown on 20 April 2022, in Zhuanghe City, Liaoning Province, transplanted on 25 May and harvested on 20 January 2023. In Baodi District, Tianjin, the sowing and transplanting times were on 20 April and 20 May 2022, respectively, while harvested occurred on 20 November 2022. In Fengyang City, Anhui Province, rice was sown from 15–20 May 2022, transplanted on 20 June and harvested on 20 November 2022. The monthly maximum temperature during the rice growing season is shown in Figure S1.

*2.3. Indicator Measurement*

2.3.1. Yield and Its Constituent Factors

Three replicates of each variety at the maturity stage in each region were selected. From each, 5 adjacent holes showing uniform growth with no missing seedlings were sampled. The number of grains per panicle, setting rate and 1000-grain weight were determined. The theoretical yield (kg/mu) of each variety was then calculated using agronomic trait data. Theoretical yield (kg/mu) = effective ear number (kg/mu) × spike grain number (grain) × seed set rate (%) × 1000 grain weight (gram) × $10^{-6}$.

2.3.2. Soft Rice Taste Quality

First, 30 g of whole rice was placed in a round aluminum can with uniform specifications in the laboratory. The rice was washed repeatedly for 30 s using a lid with a filter screen until the water ran clear. Then, 40.5 g of water was added, the container was covered with filter paper and it was sealed tightly with a high-temperature-resistant rubber band. The rice grains were soaked starting from the beginning of washing and the aluminum can with the filter paper was placed in a steamer for 30 min. After cooking, the power was turned off and the can was left in the pot for 10 min. The aluminum can was removed and the rice was stirred evenly, covered with filter paper and placed on a shelf to cool naturally for 20 min. Afterward, the filter paper was removed, the can was re-sealed with an appropriate aluminum lid for 1.5 h and finally exactly 8.00 ± 0.1 g of rice was sampled for analysis. A rice taste meter (STA1A type, SATAKE Corporation, Hiroshima, Japan) was used to determine the sample's taste value, viscosity, hardness and other related data indicators.

2.3.3. Determination of Amylose Content in Soft Rice

The AC of various rice starch types was determined using an AA3 continuous flow analyzer produced by BRAN + LUEBBE, Germany, analyzed by matching computer software AACE (Version 6.04). A rice flour sample of 0.1000 g ± 0.1005 g was weighed using an analytical balance with an accuracy of 0.0001 g. Three replicates were used for each sample and kept in 100 mL volumetric flasks. Sequentially, 1 mL anhydrous ethanol (100%) was added, the label was pasted on the table and the rice flour was mixed slowly to keep it at the bottom of the flask. Then, 9 mL of 1 N sodium hydroxide solution was added in two

portions (4.5 mL each time) and slowly shaken well to mix evenly. Each flask was sealed with plastic wrap, then five of them were sealed tightly with a high-temperature-resistant rubber band and placed in a boiling water bath for 15 min. Then, they were removed and allowed to cool on the table. After cooling, each flask was filled with distilled water up to 100 mL and shaken well. These solutions were then used as the liquid. The ACs of the four standard samples used in this study were 2%, 11%, 16% and 27%, respectively. With the AC of the standard sample as the vertical coordinate and absorbance value of the corresponding sample as the horizontal coordinate, a standard curve was prepared. The relationship is described by the following formula: $Y = a + bx$ (Y: AC of the sample; a: standard curve intercept; b: standard curve slope; x: sample absorbance value). The AC of the sample was calculated by using this standard formula.

### 2.3.4. Determination of Protein Content in Soft Rice

The national standard "ISO/TS 16634-2:2009, Food products-Determination of the total nitrogen content by combustion according to the Dumas principle andcalculation of the crude protein content- Part 2: Cereals, pulses and milled cereal products, MOD" was used. We used the nitrogen analyzer RapidNcube from Elementar Analysensysteme GmbH (Langenselbold, Germany) to determine the protein content (PC) of several rice varieties. The sample was crushed with a grinder, sieved through a 100-mesh screen, mixed and stored in a sealed bag. Using an analytical balance with an accuracy of 0.0001 g to weigh 100 mg $\pm$ 1 mg of the sample, it was placed on tin foil, and the corresponding tool was used to press until the sample powder was completely wrapped in the tin foil. Before the test, to stabilize the instrument, sucrose equivalent to the sample's weight was used to make three blanks. Subsequently, a standard curve was drawn and the standard substance aspartate was used for calibration three times. The samples to be tested were then placed in sequence.

### 2.3.5. Determination of Consistency of Soft Rice Glue

The sample glue consistency (GC) was determined according to the national standard "GB/T 22294-2008 Determination of Rice Glue Consistency". First, the sample was sieved through a 100-mesh screen. Then, 100 mg $\pm$ 1 mg of rice flour was added to a scaled test tube, 0.2 mL of 0.025% thymol blue ethanol solution was added and a vortex mixer was used to ensure the rice flour was fully dispersed. Then, 2.0 mL of 0.200 mol/L potassium hydroxide solution was added. The test tube was shaken vigorously to thoroughly mix. Then, the tube was covered and immediately placed in a boiling water bath for 8 min. The cover was removed, the test tube was placed on a rack to cool for 5 min and then it was transferred to an ice bath at 0 °C to cool for another 20 min.

After taking out the cooled test tube, it was immediately placed on a horizontal operating table adjusted with a level and left at room temperature (25 °C $\pm$ 2 °C) for 1 h. The amount of rice glue that flowed within the test tube was immediately measured. Three sets of replicates were made for each sample with the absolute value of the duplicate data results being within $\pm$7 mm.

### 2.3.6. Determination of RVA Characteristic Value of Rice Flour

The Rapid Visco Analyzer (RVA) was used for rapid viscosity analysis with supporting software TCW used for data analysis. First, 3.00 g of milled rice flour was passed through a 100-mesh screen, then 25 mL of deionized water was added to the test tank containing the rice flour and stirred well. The RVA initially stirred rapidly at 960 r/min for 10 s, then the speed was reduced to 160 r/min to maintain consistency throughout the process. Next, the rice flour paste was uniformly heated from 50 °C to 95 °C, held at this temperature for 2.5 min and then cooled to 50 °C for 1.4 min. The total duration of the procedure was 12.5 min. The characteristic values obtained from the RVA spectrum included peak viscosity (PKV), hot paste viscosity (HPV), cool paste viscosity (CPV), break-

down value (BDV = PKV − HPV), setback value (SBV = CPV − PKV), consistence value (CSV = CPV − HPV), gelatinization temperature and gelatinization time.

### 2.3.7. Determination of Amylopectin Chain Length Distribution in Soft Rice

Initially, the sample was pre-treated: 10 mg of purified starch was weighed and suspended in 5 mL of water, then placed in a boiling water bath for 60 min with intermittent vortex mixing. Next, 50 μL of sodium acetate (0.6 M, pH 4.4), 10 μL of $NaN_3$ (2% $w/v$) and 10μL of isoamylase (1400 U) were added and incubated at 37 °C for 24 h. Then, 0.5% ($w/v$) sodium borohydride solution was added, mixed vortexically and left for 20 h. Then, 600 μL was added to a centrifuge tube and evaporated to dryness under a stream of nitrogen at room temperatureIt was dissolved in 30 μL 1M NaOH for 60 min, then 570 μL of water was added to dilute and centrifuged at 12,000× $g$ rpm for 5 min and the supernatant was taken for analysis.

The starch was analyzed by high-performance anion exchange chromatography with spectral data processed by Thermo ICS5000 software. The chromatographic system used was a Thermo ICS5000+ with Dionex CarboPac PA200 (250 × 4.0 mm, 10 μm) liquid chromatography column. The inoculated sample volume was 5 μL. Mobile phase A: 0.2 M NaOH; Mobile Phase B: 0.2. M NaOH/0.2 M NaAC. The column temperature was maintained at 30 °C and components were detected by an electrochemical detector [11–13]. The flow rate was set at 0.4 mL/min. The elution gradient was as follows: 0 min A/B (90:10 $v/v$), 10 min A/B (90:10 $v/v$), 30 min A/B (40:60 $v/v$), 50 min A/B (40:60 $v/v$), 50.1 min A/B (90:10 $v/v$) and 60 min A/B (90:10 $v/v$). Peaks corresponding to each degree of polymerization (DP) were identified and quantified [11–13].

### 2.3.8. Determination of Molecular Weight Distribution of Soft Rice Starch

The samples were pre-treated by weighing 5 mg of purified starch and dissolving it in 5 mL DMSO as mobile phase at 80 °C for 3 h. The chromatographic system used was a gel chromatograph difference multi-angle laser light scattering system. The liquid chromatography system was a U3000 (Thermo, Waltham, NA, USA), differential refractometer was an Optilab T-rEX (Wyatt technology, Goleta, CA, USA) and light scattering detector was DAWN HELEOS II (Wyatt technology, Goleta, CA, USA), using a wavelength of 663.7 nm.

For separation, gel exclusion columns suitable for the molecular weight range were used: Ohpak SB-805 HQ, SB-804 HQ and SB-803 HQ (all 300 × 8 mm). Column temperature was maintained at 60 °C, with an insertion volume of 200 μL. Mobile Phase A consisted of 0.5% LiBr in DMSO. The flow rate was set at 0.3 mL/min with an elution gradient maintained isodegree for 120 min. The DMSO solution had a dn/dc value of 0.07 mL/g. Chromatographic data were processed by ASTRA6.1 software and the absolute molecular weight analysis diagram plotted time (min) as the horizontal coordinate and molar mass (g/mol) as the vertical coordinate [14–17].

### 2.4. Statistical Analysis

Statistics and analyze raw data (Microsoft, USA), Analysis of variance and correlation analysis were performed with SPSS26.0 software (SPSS corporation, Chicago, IL, USA), Two-way analysis of variance (ANOVA) and means were compared by Tukey's test at the 5% level, Graphed with Origin 2021 (OriginLab corporation, Northampton, MA, USA).

## 3. Results

### 3.1. Changes in Soft Rice Yield and Its Constituent Factors in Different Regions

To assess the performance of yield components of a single variety of soft rice planted in different regions (Table 1), we compared the data from Anhui with other areas. In the Liaoning region, DGR1 and Jinong Xiangruan 1 showed increases of 24% and 8% in grains per panicle, respectively. Seed setting rates increased by 29% for DGR1 and 11% for Jinong Xiangruan 1, while the 1000-grain weight increased by 4% and 5%. The theoretical yield for these varieties increased significantly by 72% and 53%, respectively. Additionally,

the effective number of panicles for Jinong Xiangruan 1 in Liaoning was higher by 22%, reaching statistical significance. In comparison, in the Tianjin area, the number of grains per panicle of DGR1 increased by 24%, while for Jinong Xiangruan 1 it decreased by 14%. Seed setting rates were 28% for DGR1 and 13% for Jinong Xiangruan 1. The effective panicle number and theoretical yield of DGR1 increased by 16% and 96%, respectively, with these differences also being statistically significant. The 1000-grain weight across different areas showed no significant variation.

**Table 1.** Yield and its constituent factors of soft rice varieties in different regions.

| Region | Effective Panicles | | Grains per Spike | | Setting Rate (%) | | 1000-Grain Weight (g) | | Theoretical Yield (kg/M) | |
|---|---|---|---|---|---|---|---|---|---|---|
| | DGR1 | JNXR1 | DGR1 | JNXR1 | DGR1 | JNXR1 | DGR1 | JNXR1 | DGR1 | JNXR1 |
| Liaoning | 20 b | 22 a | 114 a | 123 a | 87 a | 93 a | 25.5 a | 25.7 a | 701.60 b | 895.66 a |
| Tianjin | 22 a | 17 b | 114 a | 98 c | 88 a | 94 a | 26.2 a | 25.2 a | 800.96 a | 547.67 b |
| Anhui | 19 b | 18 b | 92 b | 114 b | 69 b | 83 b | 24.6 a | 24.6 a | 409.18 c | 584.54 b |

Data are means ± SE (*n* = 3). Different letters indicate significant differences between treatments according to two-way ANOVA followed by Tukey's test (*p* < 0.05).

### 3.2. Differences in Eating Quality of Soft Rice of the Same Variety in Different Regions

We analyzed the physical and chemical indices of the cooking and taste quality of various soft rice varieties across different regions (Figure 1). In all the regions except Anhui, glue consistency, PC and rice taste value showed no significant differences. However, the AC of the two soft rice varieties exhibited significant regional variation. Data screening revealed that the varieties sown in Tianjin and Liaoning shared similar food taste values and PC levels but differed in AC.

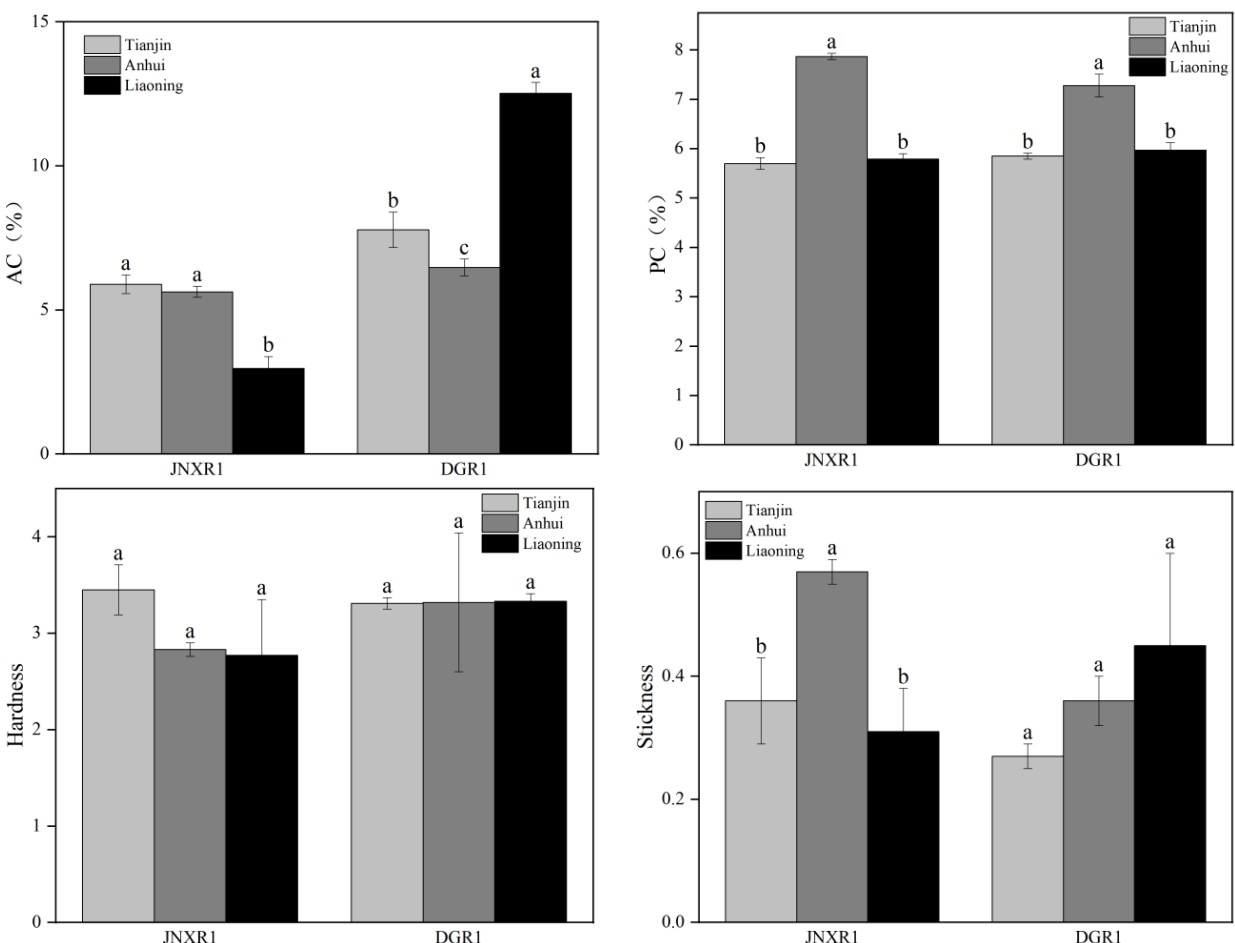

**Figure 1.** *Cont.*

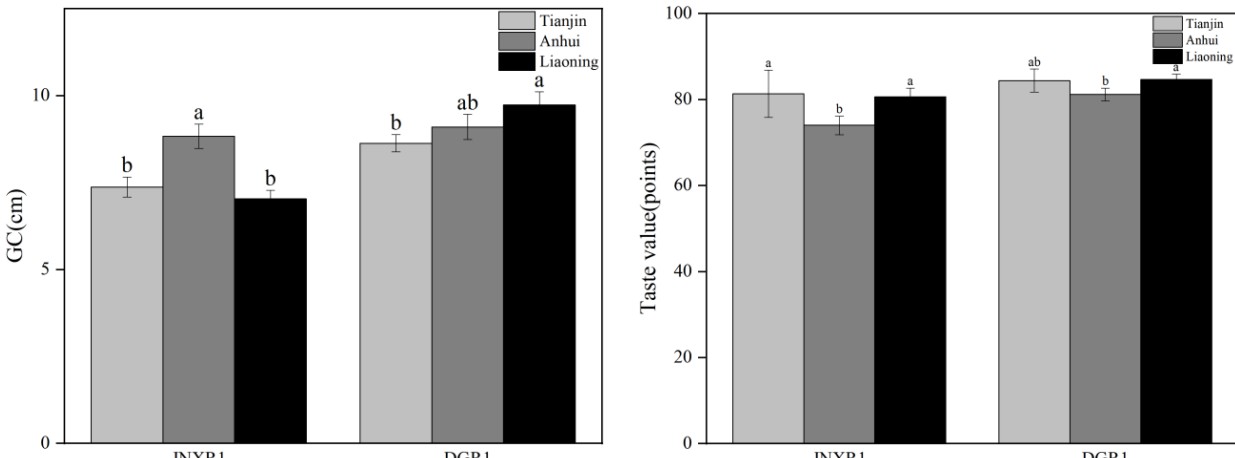

**Figure 1.** Physicochemical indicators of taste quality for soft rice varieties in different regions. Data are means $\pm$ SE (*n* = 3). Different letters indicate significant differences between treatments according to two-way ANOVA followed by Tukey's test (*p* < 0.05).

Previous research has shown that amylose content has a significant effect on the hardness and viscosity of rice; that is, the higher the amylose content, the harder the rice; the lower the amylose content, the greater the viscosity of rice [18]. Despite a uniform cooking process across all varieties, significant differences in hardness and viscosity were observed. Soft rice varieties from regions with higher taste values revealed increased viscosity compared to those with lower taste values, although their hardness was reduced.

### 3.3. Differences in Viscosity Characteristics of Soft Rice of the Same Variety in Different Regions

The RVA spectrum of rice starch effectively simulates the cooking process, making RVA characteristic values important indicators of rice's cooking and taste quality. These values are also essential for the selection and breeding of high-quality rice varieties [19]. RVA spectral characteristic values include peak viscosity (PKV), hot paste viscosity (HPV), cool paste viscosity (CPV), breakdown viscosity (BDV = PKV − HPV), setback viscosity (SBV = CPV − PKV), consistence viscosity (CSV = CPV − HPV) and peak temperature (PaT °C). PKV can reflect the expansion degree of starch and water binding capacity; HPV is the shear resistance of starch after high temperatures; CPV refers to the regeneration capacity of rice pulp after high temperatures, which is related to the gel capacity of starch formed after cooking and cooling; BDV is used to measure the difficulty of destroying the expanded starch particles, which indicates the degree of stability of the rice during the cooking process. Significant differences in RVA characteristic curves exist among the same varieties of soft rice sown in different regions (Figure 2). Table 2 presents the RVA characteristic values for various soft rice varieties across different regions. The data indicate that, compared to the Tianjin region, the DGR1 in Liaoning shows higher setback viscosity, final viscosity and setback value, while its peak viscosity and breakdown value are lower. These observations might initially suggest different AC between these regions. Typically, higher setback and final viscosities could indicate a higher AC in Liaoning. However, the lower peak viscosity and breakdown values in Liaoning suggest a lower AC. This obvious contradiction may require further investigation or additional data to clarify the AC differences between the Tianjin and Liaoning regions.

Additionally, the RVA characteristic values such as peak viscosity and minimum viscosity of Jinong Xiangruan 1 are lower in Liaoning compared to Tianjin, where its AC is higher. This difference in AC, with Liaoning tending towards the lower levels typical of glutinous rice (3%), influences the RVA characteristics. Previous studies confirm that glutinous rice generally has lower RVA values, explaining why the values for Jinong Xiangruan 1 in Liaoning are also lower. Consequently, the RVA values of DGR1 and

Jinong Xiangruan 1 are not directly comparable across these regions due to these complete differences in their starch properties.

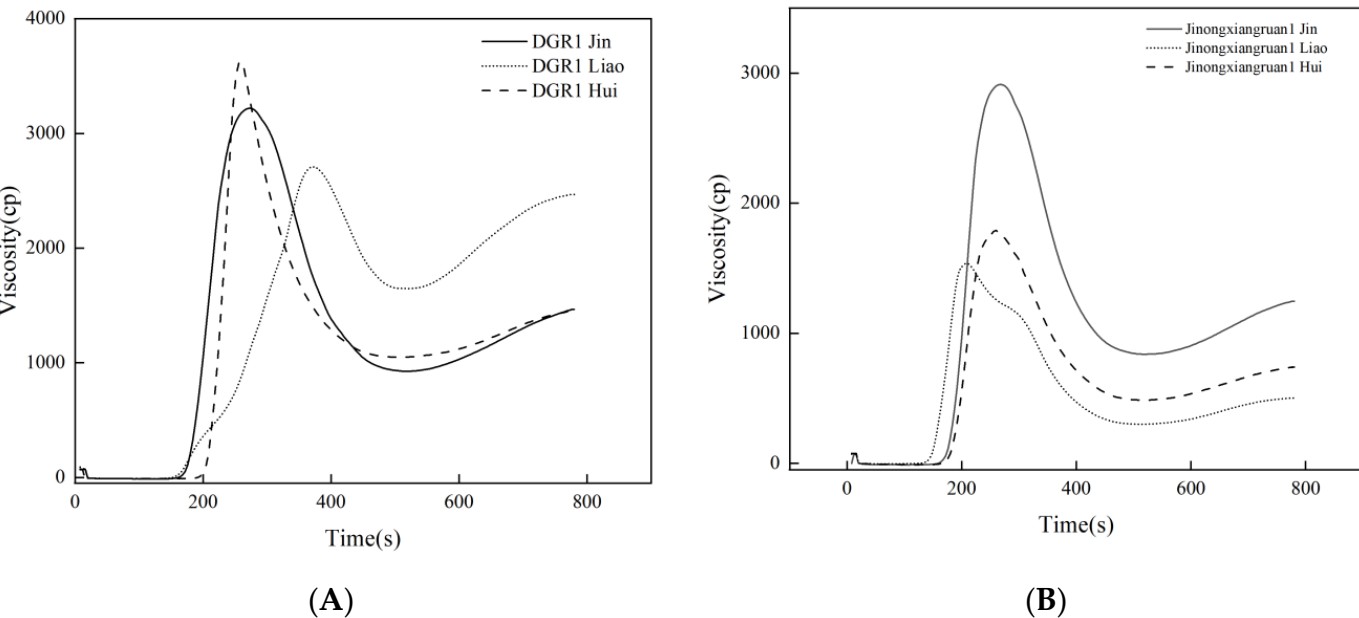

**Figure 2.** RVA profiles of two soft rice varieties in different regions. The influence of northern and southern regions on the viscosity characteristics of soft rice planted with the same variety was explored. (**A**,**B**) show the RVA spectrum comparison curves of DGR1 and Jinong Xiangruan 1 in different regions.

**Table 2.** RVA characteristic values of starch in soft rice varieties from different regions. Data are means ± SE (*n* = 9). Different letters indicate significant differences between treatments according to two-way ANOVA followed by Tukey's test (*p* < 0.05).

| Sample | Region | PKV (cp) | HPV (cp) | CPV (cp) | BDV (cp) | CSV (cp) | PaT (°C) |
|--------|--------|----------|----------|----------|----------|----------|----------|
| | Tianjin | 3232.5 ± 41.3 b | 952.3 ± 22.5 b | 1490.9 ± 32.7 b | 2233.1 ± 49.0 b | 538.4 ± 10.3 b | 71.1 ± 0.1 b |
| DGR1 | Anhui | 3628.0 ± 9.6 a | 1043.3 ± 4.2 b | 1441.7 ± 11.6 b | 2584.7 ± 12.0 a | 398.3 ± 7.5 c | 76.9 ± 0.5 a |
| | Liaoning | 2683.6 ± 65.8 c | 1710.3 ± 86.0 a | 2528.7 ± 75.3 a | 973.3 ± 77.7 c | 818.3 ± 11.9 a | 70.4 ± 0.5 b |
| | Tianjin | 3004.2 ± 137.4 a | 870.8 ± 14.9 a | 1259.3 ± 5.0 a | 2133.4 ± 137.9 a | 397.0 ± 10.4 a | 71.4 ± 0.5 b |
| JNXR1 | Anhui | 1849.7 ± 63.4 b | 505.0 ± 22.6 b | 764.3 ± 27.7 b | 1344.7 ± 41.1 b | 259.3 ± 5.1 b | 72.5 ± 0.1 a |
| | Liaoning | 1655.7 ± 101.0 b | 321.7 ± 23.0 c | 531.0 ± 32.4 c | 1334.0 ± 85.2 b | 209.3 ± 9.5 c | 66.3 ± 0.5 c |

Data are means ± SE (*n* = 3). Different letters in the same column indicate significant differences between treatments according to two-way ANOVA followed by Tukey's test (*p* < 0.05).

Experimental results further illustrate that in soft rice varieties like DGR1, higher AC correlates with increased minimum viscosity, final viscosity and setback value. This relationship is due to the recombination of amylose molecules during cooling, which also affects the setback value. The setback value measures the degree of recrystallization process of gelatinized starch. Conversely, a lower AC is associated with higher peak viscosity (HPV) and breakdown value, reflecting the maximum viscosity achieved during starch gelatinization in water heating and the subsequent structural breakdown of gelatinized starch particles.

### 3.4. Absolute Molecular Weight Distribution of Soft Rice Starch

The starch characteristics of soft rice varieties with similar taste values and PC but differing significantly in AC were analyzed. The absolute molecular weight distribution of total starch, amylose and amylopectin was assessed using gel chromatography (SEC-MALLS-RI) (Figure 3). In the DGR1 from Tianjin, the polydispersion index (a measure of molecular weight distribution) for amylopectin and total starch indicates that values are

greater in Liaoning than in Tianjin. Meanwhile, the polydispersion coefficient for amylose is lower in Liaoning than in Tianjin. In contrast, for Jinong Xiangruan 1, both the amylose and total starch polydispersion coefficients are lower in Liaoning compared to Tianjin, whereas the polydispersion index for coefficients for amylopectin is higher in Liaoning.

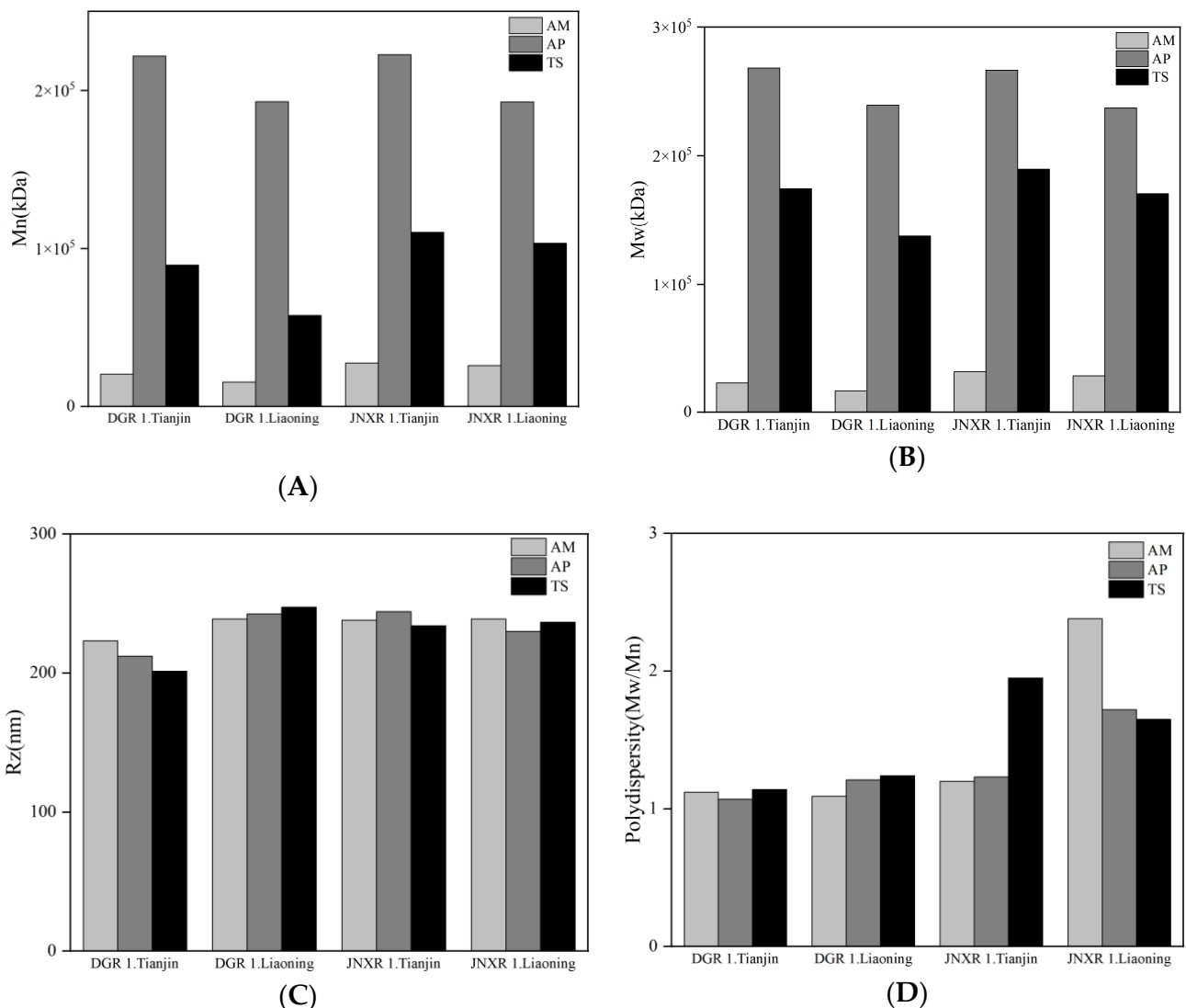

**Figure 3.** Comparison of characteristic indexes of relative molecular weight distribution of starch in each sample. The average molecular weight distribution of starch number (**A**), weight average molecular weight distribution (**B**), Z-average rotational radius (**C**) and starch polydispersion coefficient ($M_w/M_n$) (**D**) of each sample were compared. Among them, AM, AP and TS represent amylose, amylopectin and total starch, respectively.

Additionally, we explored the relationships between the molecular weights of amylose and amylopectin across these varieties. We determined the ratio of the number-average molecular weight ($M_n$) to the weight-average molecular weight ($M_w$) for amylose and amylopectin of soft rice varieties. The ratio of $M_n$ to $M_w$ for amylopectin in DGR1 is higher than that in Liaoning (Figure 4). For Jinong Xiangruan 1, this ratio indicates that Tianjin has a lower molecular weight distribution compared to Liaoning.

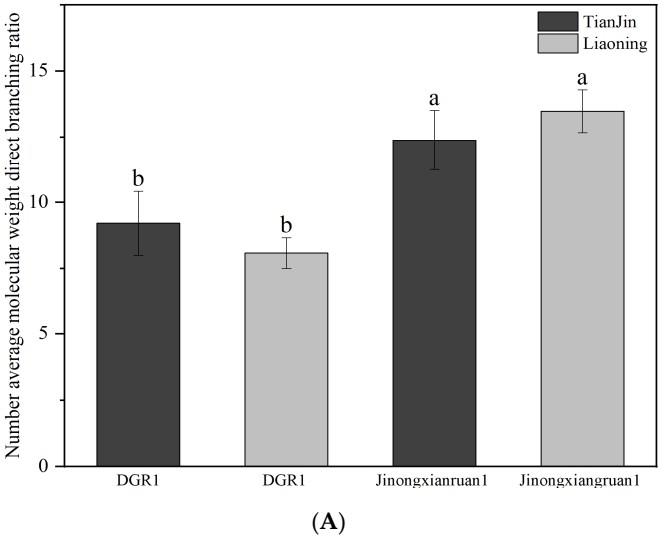

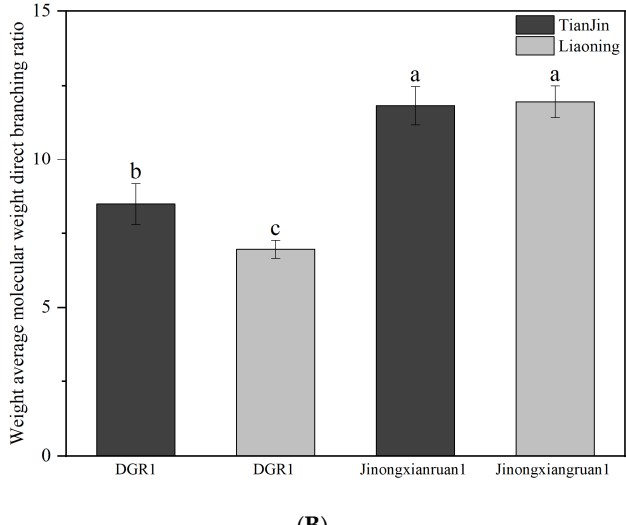

**(A)**

**(B)**

**Figure 4.** Ratio of number-average molecular weight to weight-average molecular weight for amylose and amylopectin in two types of soft rice varieties (**A**,**B**). Data are means ± SE (*n* = 3). Different letters indicate significant differences between treatments according to two-way ANOVA followed by Tukey's test (*p* < 0.05).

### 3.5. Distribution of Amylopectin Chain Length in Soft Rice

The 1,6-glucoside bonds in amylopectin were hydrolyzed by isoamylase, resulting in the formation of dextran chains of varying lengths. These chains were then quantified according to their degree of polymerization using ICS ion chromatography (Figure 5A,B). The amylopectin chain length distributions for the two studied varieties showed a similar bimodal distribution, with the highest peak value at DP12. The content at this DP was higher in Liaoning than in Tianjin.

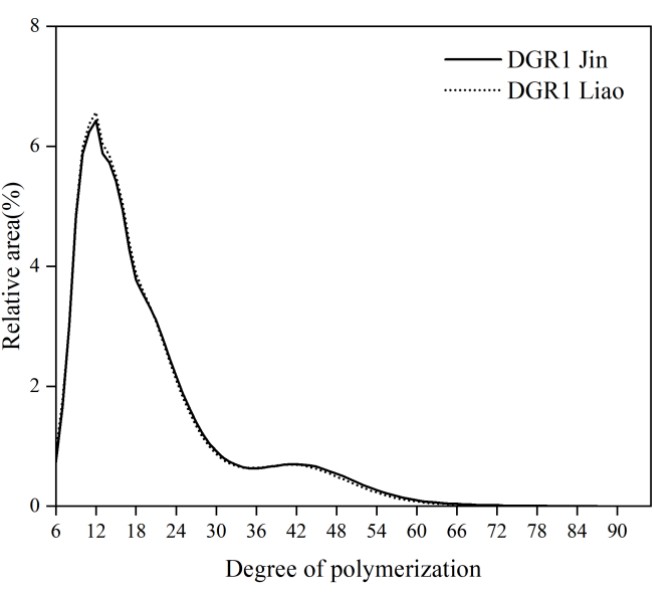

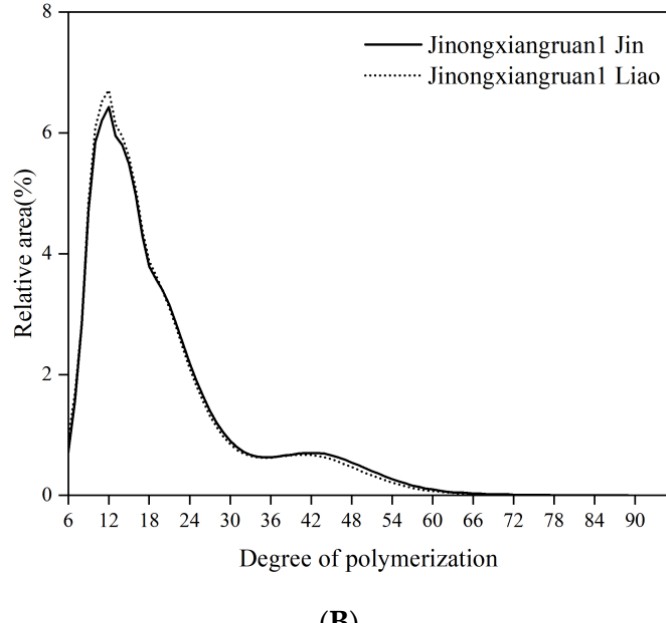

**(A)**

**(B)**

**Figure 5.** Distribution of branched chain length in starch of DGR1 and Jinong Xiangruan 1 rice in Liaoning and Tianjin regions (**A**,**B**).

To further explore the variations in amylopectin chain length distribution among rice varieties that have similar taste values but vary in AC across regions, the experimental data

were analyzed. Comparative analysis generated relative chain length distribution diagrams for two groups of soft rice (Figure 6A,B). For DGR1, the content of short-chain A and medium-short-chain B1 (DP6-20) was higher in Liaoning compared to Tianjin. Conversely, the content of medium-long-chain B2 and long-chain B3 was lower in Liaoning than in Tianjin. Similarly, for Jinong Xiangruan 1, the content of short-chain A and medium-short-chain B1 (DP6-19) was higher in Liaoning compared to Tianjin.

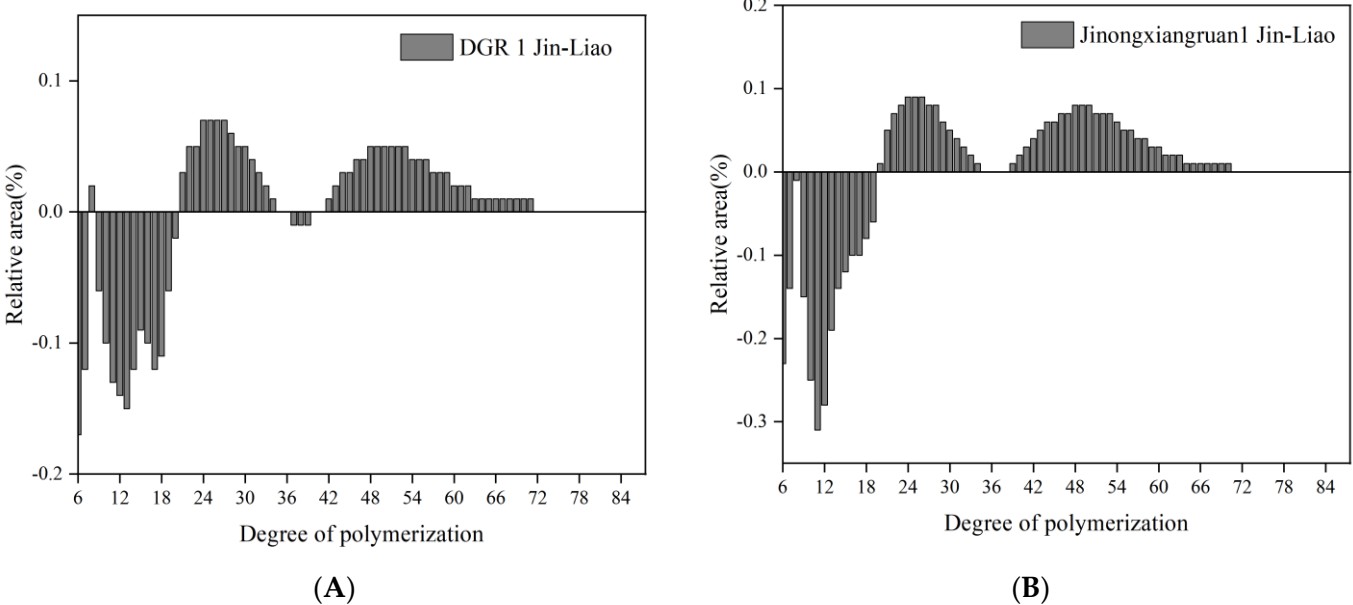

**(A)**       **(B)**

**Figure 6.** Comparison of the distribution of relative chain length distribution of branched chain starch in DGR1 (**A**) and Jinong Xiangruan 1 (**B**) in Liaoning and Tianjin regions. The light–colored and dark–colored accumulations represent the relative chain length distribution of amylopectin in Liaoning and Tianjin regions, respectively.

*3.6. Correlation Analysis of Absolute Molecular Weight Distribution and Chain Length Distribution of Amylopectin*

The $M_n$ is significantly negatively correlated with the A chain of amylopectin and positively correlated with average chain length (Figure 7). Additionally, $M_n$ has a negative correlation with the B1 chain and a positive correlation with the branched chain length of B2 and B3; however, these correlations are not significant. The $M_w$ is positively correlated with the B3 chain and average chain length and negatively with the short branched chains A and B1, while being positively correlated with the long branched chain B2, though these correlations do not reach significance. The z-average molecular weight ($R_z$) shows a significantly negative correlation with the short branched chain B1 and positive correlation with long branched chain B3. It is negatively correlated with short branched chain A and positively with the long branched chain B2, though these correlations are not significant. The polydispersity index ($M_w/M_n$) is significantly positively correlated with amylopectin's short chain A and negatively with the long chain B2 and average chain length. It shows a positive correlation with the short chain B1 of the branched chain and negative correlation with the long chain B3, although these correlations are not significant. Additionally, there is a significant negative correlation between the short chain A and the average branch chain length.

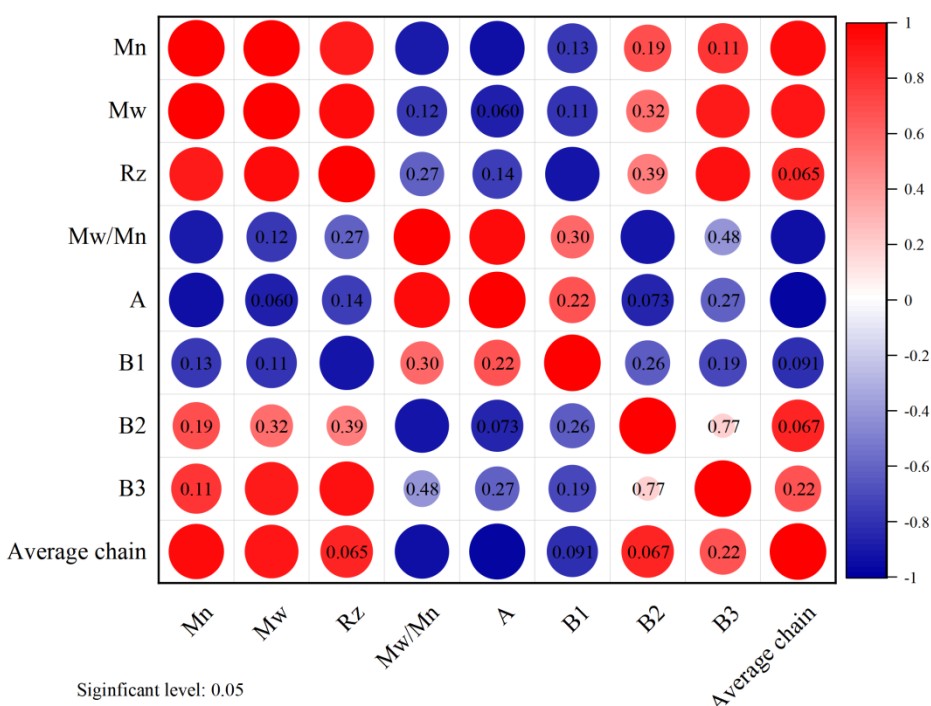

**Figure 7.** Correlation between absolute molecular weight distribution and chain length distribution of amylopectin.

## 4. Discussions

### 4.1. Effects of Differences in Temperature and Light Conditions in North and South China on Yield, Components and Cooking Quality of Soft Rice of the Same Variety

Ecological and environmental conditions during the growth and development of rice play a key role in determining its yield and quality [7]. Photosynthetic assimilates during the filling and fruiting stages of rice are vital for the processes of grain filling and accumulation. Thus, abnormal temperature and light conditions can inhibit the grain filling process in rice, thereby significantly reducing the grain setting rate and 1000-grain weight [20]. Experimental results revealed that the average monthly maximum temperature during the filling to fruiting stages in Anhui Province reached 40.3 °C (Figure S1), significantly higher by 10–15 °C compared to Liaoning and Tianjin. This excessive maximum temperature likely shortened the filling duration, thereby reducing the time for grains to gain their maximum weight. As a result, the seed weight of soft rice varieties in Anhui was significantly lower than that in Tianjin and Liaoning (Table 1).

A rice taste meter evaluates the taste value of rice after being crushed or cooked [21]. This taste value reflects cooking standards for cooked rice. The results revealed that the taste value of soft rice varieties in Anhui was significantly lower than those in Liaoning and Tianjin, where the hardness of rice was also significantly higher (Figure 1). Both AC and PC are key factors influencing the taste quality of rice. Research suggests that within a certain range, reducing AC and PC can effectively decrease the hardness and increase the viscosity, thereby enhancing the taste quality of rice [22]. Some studies have revealed that an increase in AC correlates with increased rice hardness, negatively affecting the taste value [23]. However, our study's results do not align with this finding. In this experiment, although the AC levels in the soft rice varieties from Liaoning and Tianjin were significantly different, no significant differences in taste value or rice hardness were observed, likely due to regional environmental influences. Some studies reveal that high temperature can increase the AC of rice [24], while other studies find that high temperature conditions can lead to a decrease in AC of rice in the grain setting stage [25]. Therefore, there is no conclusive study on the relationship between the grain setting stage and AC at present. In this study, the AC of each soft rice variety in Anhui was relatively low, yet the PC was high.

Despite poor performance compared to the other regions, the soft rice in Anhui showed a higher food taste value [26]. It can be seen that if the PC or AC of rice is too high or too low, it will lead to the deterioration of rice's cooking taste quality.

### 4.2. Relationship between Gelatinization Characteristics and Cooking Taste Quality Indexes of the Same Soft Rice Varieties in North and South China

The viscosity characteristics of rice starch, an important index reflecting cooking and eating quality, have received attention. Viscosity analyzers are gradually being used to measure rice taste quality [27,28]. Some studies indicate that the RVA spectrum of rice is closely related to cooking and eating quality; particularly, RVA characteristic values provide a good reflection of these qualities [29]. The experimental results revealed that the RVA eigenvalues for the same variety of soft rice from the northern and southern regions exhibited significant differences (Table 2). The test variety DGR1 showed relatively low AC and relatively high PKV and BDV [30]. Some studies suggest that increased PKV and BDV can occur under high-temperature conditions [31], which is consistent with our findings in the Anhui region, where temperatures are higher than in the other studied regions. In the experimental variety Jinong Xiangruan 1, the RVA characteristic values such as PKV and BDV, which are associated with high AC, exhibited relatively high levels, differing from the results of previous studies. The reason may be that Jinong Xiangruan 1 sown in Anhui and Liaoning regions forms glutinous rice during its growth and development due to environmental factors. Consequently, the RVA characteristic values of glutinous rice are low.

Additionally, some studies have found that high-temperature conditions can increase the gelatinization temperature of rice [32] and result in decreased or negligible changes in adhesive consistency [33]. In this study, the test materials in Anhui presented high gelatinization temperatures (Table 2), consistent with previous findings. However, the glue consistency is also relatively high, which is consistent with research generally showing that AC is negatively correlated with glue consistency [33]. During starch heating, the long chains in amylopectin, with a proportion similar to that of amylose, can form spiral complexes with lipids that are not easily disconnected. This maintains the integrity of starch particles during heating, resulting in lower PKV and BDVs [34].

### 4.3. Effects of Synergistic Effects of Fine Structure and Physicochemical Properties of Starch on the Taste Quality of Soft Rice

Studies have shown that in addition to AC and PC, the fine structure of starch also plays a vital role in the formation of rice quality [35]. While the ACs of some rice varieties may be similar, significant differences in their taste value can be attributed to variations in the fine structure of amylopectin [2]. This study revealed that the two soft rice varieties in Tianjin and Liaoning exhibited similar taste values and PCs but significantly different ACs. The ratio of straight chains to branched chains and the distribution of chain lengths in rice starch directly affects the molecular weight distribution of starch [36], which in turns influences its physicochemical properties. Analysis of the $M_w$ and chain length distribution of amylopectin revealed that $M_w$ reflects the starch DP; a higher $M_w$ indicates a greater DP of amylopectin. Results show that in Tianjin, both DGR1 and Jinong Xiangruan 1 have a higher $M_w$ than in Liaoning, indicating a lower average DP and thus a lower average $M_w$ in Tianjin (Figure 3). $R_z$ is usually used as an indicator of the branching degree of starch [33]. The higher $R_z$ value indicates a greater degree of branching. In this study, the $R_z$ value of amylose with high AC in DGR1 was lower, while the $R_z$ values for amylopectin and total starch in $M_w$ were higher, suggesting that amylose has a lower degree of branching and that there is a higher degree of branch aggregation [37]. The starch from DGR1 in Liaoning exhibited a larger $M_w$ and higher density. In contrast, Jinong Xiangruan 1 with high AC showed lower $R_z$ values for straight chains, branched chains and total starch $M_w$. $M_w / M_n$, the polydispersion coefficient, is used to show the distribution of starch molecules [27,38]. In this study, the two soft rice varieties showed that those with higher AC had a smaller $M_w / M_n$ value, indicating a larger molecular weight span, a lower degree of dispersity

and more dispersed molecular weight size; this suggests a non-uniform molecular mass distribution [39]. Despite similar taste values and PCs, the rice with higher AC exhibited more amylopectin and a larger number of branches. By analyzing the distribution of amylopectin chain lengths, it was observed that the chain length distribution curves of the same varieties in the two regions were mostly similar, with the position of the first peak indicating the short chains and the second peak indicating the long chains of the branches (Figure 5). Chains A and B1 had the largest proportion of total amylopectin (Figure 8). In DGR1, those with higher AC had a higher content of A chains and B1 chains in amylopectin. Therefore, it is believed that the reason why the same soft rice in different regions can maintain similar taste values is due to the balance of high AC, high content and proportion of amylopectin short chains and relatively dispersed $M_w$. The contrasting findings in Jinong Xiangruan 1 might be due to the AC in Liaoning being as low as 3%, which aligns with the amylose range of glutinous rice. This results in a larger proportion and higher content of short amylose chains, a greater degree of branching and more concentrated molecular weight distribution under similar taste conditions.

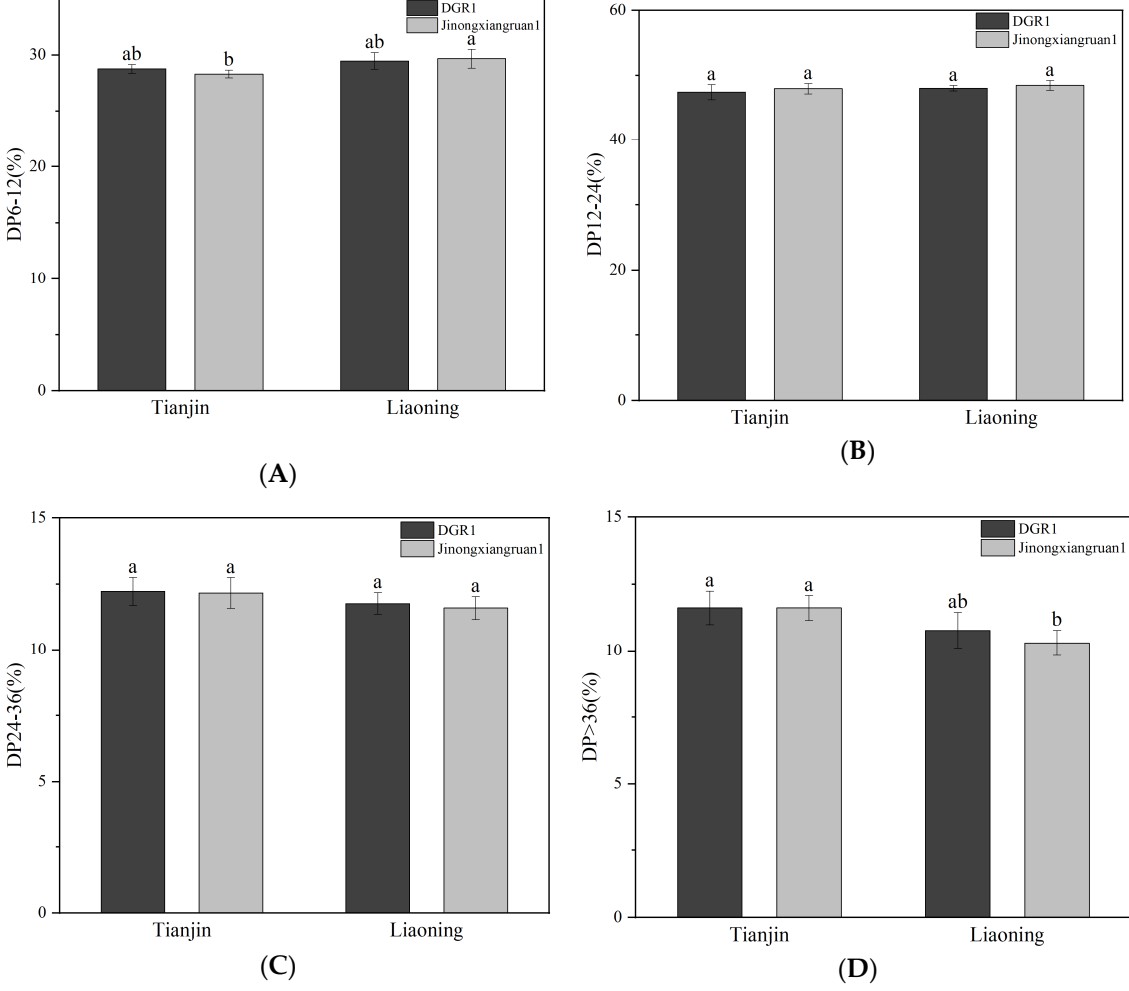

**Figure 8.** *Cont.*

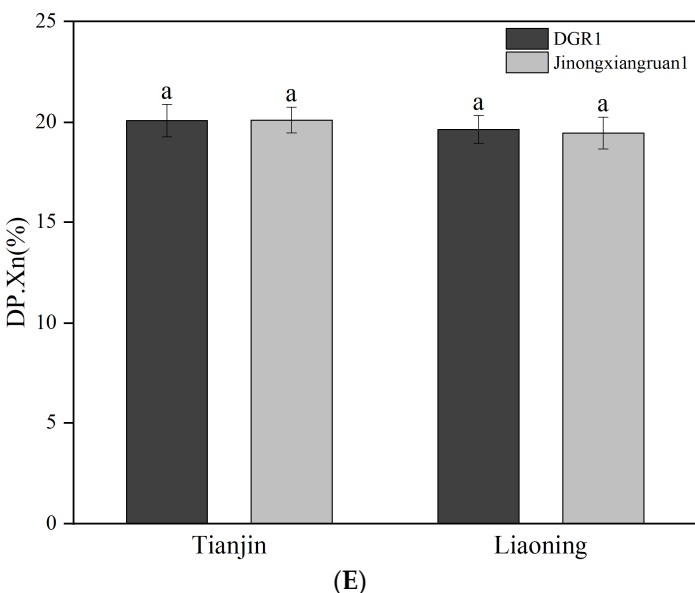

**Figure 8.** The branched chain length structure of amylopectin was determined. DP is the degree of polymerization, DP6-12% is the content of short chain A in amylopectin (**A**). DP12-24% is the content of short chain B1 in amylopectin (**B**). DP24-36% is the content of long chain B2 in amylopectin (**C**). DP > 36% is the content of long chain B3 chain of amylopectin (**D**). DP.Xn (%) represents the average chain length content of amylopectin (**E**). Data are means ± SE ($n = 3$). Different letters indicate significant differences between treatments according to two-way ANOVA followed by Tukey's test ($p < 0.05$).

## 5. Conclusions

This study analyzed the physical, chemical indices, as well as the fine structure of starch, in the same soft rice varieties sown in different regions, which exhibited similar taste values and PC but substantial differences in AC. The analysis using SEC-MALLS-RI showed that the soft rice with low AC exhibited a relatively high direct branch ratio, number-average molecular weight and $M_w$ of both amylose and amylopectin, resulting in a relatively concentrated total starch $M_w$. The ICS analysis showed that when the AC was 5–12%, the content of short branched chains (DP6-12) in soft rice with high AC was lower, while the content of short branched chains (DP12-24) was higher. These structural variations may lead to higher PKV and BDV, alongside lower cold adhesive viscosity and CSV. These findings underscore that changes in the fine structure of starch significantly influence the taste quality of the same variety of soft rice under different regional conditions. Moreover, this study suggests that the absolute molecular weight indices of amylose and amylopectin may be the key factors in determining the amylose content across different soft rice varieties. This research provides a theoretical and technical basis for enhancing the quality of soft rice and breeding new varieties that are widely adaptable and of high quality.

**Supplementary Materials:** The following supporting information can be downloaded at: https://www.mdpi.com/article/10.3390/agronomy14051074/s1, Figure S1: The changes in minimum temperature and maximum temperature in the key period of rice growth in each region, and the average monthly maximum temperature in each region during the growth stages of rice which are, from left to right, young panicle differentiation stage, rice flowering stage, filling stage to maturity stage and rice harvest stage.

**Author Contributions:** Z.H.: Research scheme design (equal); formal analysis (equal); investigation (equal); methodology (equal); writing—original draft (lead). From Department of Agriculture and Resources and Environment, Tianjin Agricultural University, Tianjin, China. Z.I.: Formal analysis (equal); methodology (equal); writing—revision (lead). From Yangzhou University. Y.H.: Formal analysis (equal); investigation (equal); methodology (equal). From Department of Agriculture

and Resources and Environment, Tianjin Agricultural University, Tianjin, China. C.W.: Formal analysis (equal); investigation (equal); methodology (equal). From Department of Agriculture and Resources and Environment, Tianjin Agricultural University, Tianjin, China. Z.P.: Investigation (equal); methodology (equal); supervision (equal); writing—review and editing (equal). From Department of Agriculture and Resources and Environment, Tianjin Agricultural University, Tianjin, China. X.Z.: Investigation (equal); methodology (equal); supervision (equal); writing—review and editing (equal). From Department of Agriculture and Resources and Environment, Tianjin Agricultural University, Tianjin, China. J.S.: Investigation (equal); methodology (equal); supervision (equal). From Enterprises Key Laboratory of Hybrid Japonica Rice, Tianjin, China. M.Q.: Formal analysis (equal); methodology (equal); supervision (equal). From Yangzhou University. Z.L.: Director of the research (lead); research scheme design (lead); formal analysis (equal); investigation (equal); methodology (equal); writing—review and editing (equal). From Department of Agriculture and Resources and Environment, Tianjin Agricultural University, Tianjin, China. All authors have read and agreed to the published version of the manuscript.

**Funding:** This work was supported by the National Natural Science Foundation of China (32170245), Tianjin Science and Technology Planning Project (22YFZCSN00110).

**Data Availability Statement:** The data that support the findings of this study are available on request from the corresponding author. From: Department of Agriculture and Resources and Environment, Tianjin Agricultural University, Tianjin, China.

**Acknowledgments:** We acknowledge the Tianjin Agricultural University, Yangzhou University, Tianjin Enterprises Key Laboratory of Hybrid Japonica Rice.

**Conflicts of Interest:** The authors declare no conflicts of interest.

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
