# Peer review of "Physicochemical Properties and Fine Structure of Starch in Jinong Xiangruan 1 and DGR1 Soft Rice Varieties Cultivated in Different Regions of China"

_agronomy, doi:10.3390/agronomy14051074_

Round 1

Reviewer 1 Report

Comments and Suggestions for Authors

Comments on agronomy-2978887-peer-review-v1

The manuscript illustrates the physicochemical properties and fine structure of starch in Jinong Xiangruan 1 and DGR1 soft rice varieties cultivated in three different regions of China. Please use clear and concise language to convey complex scientific concepts effectively. I would like to suggest using short sentences instead of long ones for better readability and comprehension. There are several grammatical errors that need to be corrected throughout the manuscript. However, the manuscript requires the following corrections/modifications/incorporations to improve its quality -

ï‚· How do the environmental conditions in the three different ecological regions (Baodi District, Tianjin City, Liaoning Province, and Fengyang City, Anhui Province) influence the physicochemical properties and taste quality of the soft rice varieties.

ï‚· Line 8 -12: Please concise and rewrite.

ï‚· Please write the full name with its abbreviated form in the first appearance and then use only abbreviated form throughout the manuscript.

ï‚· Line 64 – Who is Reddy!?

ï‚· Line 84 – Rewrite the sentence with mentioning proper names of the city/district/province.

ï‚· Bar width can be reduced in cases of figure 4 and 8. Please check about the standard errors also, there was no error bars in the figures.

ï‚· Please use the percentage values in round figures (such as 26.8% as 27%, 11.2% as 11%, 71.5% as 72% etc.). Please check and resolve this issue throughout the manuscript.

ï‚· There are certain assumptions that need to be checked for the validity of one-way ANOVA in any experimental data sets which includes normality, homogeneity of variance and outliers. The authors should check how they may impact the interpretation of results for their experiments to continue the validity of statistical analysis.

ï‚· Please provide a detailed interpretation of the statistical findings, including the meaning of p-values, confidence intervals, effect sizes etc. Discuss the practical significance of the results in addition to their statistical significance.

ï‚· Please clarify that how the variations in temperature and light conditions affect the specific components of rice yield and quality, such as seed setting rate and 1000-grain weight? Is there any explanation for the discrepancy in the relationship between amylose content (AC) and rice hardness observed in this study compared to previous research?

ï‚· What environmental or genetic factors might contribute to the unexpected results regarding RVA characteristic values for Jinongxiangruan1 in Anhui and Liaoning regions?

ï‚· Please explain in simpler terms how the fine structure of starch, particularly the distribution of amylopectin chain lengths, influences rice taste quality? What specific characteristics of starch fine structure contribute to maintaining similar taste values in soft rice varieties from different regions?

ï‚· Please provide a brief explanation of the potential implications of this research for rice production and breeding efforts, particularly in terms of improving rice eating quality and developing new varieties?

Thus, the manuscript requires above corrections/modifications/incorporations to improve its quality that need to be addressed before its publication in the journal - Agronomy (ISSN 2073-4395) [Section: Plant-Crop Biology and Biochemistry].

Comments on the Quality of English Language

The language quality should be improved..

Author Response

Thank you very much for your comments and suggestions on the article! I will reply to your content in the form of word file.

Reviewer 2 Report

Comments and Suggestions for Authors

In this manuscript, the authors report a comparative analysis of the starch content/composition and other physicochemical, yield and taste characteristics of the grain of two soft rice varieties grown in three different climatic regions of China.

Abstract. Firstly, it is not clear on what basis the difference in amylose content is “obvious” (Line 11). Secondly, when talking about amylose, the authors write about the degree of branching (Line 14), although amylose is a linear molecule. The abstract is incomprehensible; the results in abstract are not clear and presented very confusingly, interspersed with phrases that could have been in the introduction.

Unlike the Abstract, in the Conclusion the authors provide specific results and conclusions, and this can be used to revise the abstract.

In the Introduction, the authors explain quite well how starch composition can influence the texture and flavor of a rice grain, thus justifying their goal of characterizing the characteristics (including amylose content and amylopectin fine structure) of two high-quality soft rice cultivars suitable for cultivation in different temperature zones.

Materials and methods are given in sufficient detail.

Line 92-93: “lighting duration, climatic conditions, and temperature”.  – “climatic conditions, such as photoperiod and temperature”.

Line 100-101: “The monthly maximum temperature during the rice growing season is shown in Figure 1.” Figure 1 contains “physicochemical indicators of taste quality for soft rice varieties in different regions”, but not temperature conditions. Please clarify. Probably, this is supplementary Figure S1? It would be better to indicate the monthly maximum temperature values by region in the text, and move Figure 1 to Results (para 3.2).

The results obtained are fully discussed. In particular, the authors proposed a criterion for determining the preferred qualities of grain. Such research may contribute to the development of rice germplasm evaluation criteria for the breeding of high-quality temperature-adaptive rice varieties that produce grain with good flavor characteristics.

Minor:

Line 344-345: “Photocontract products…” What is a photocontract? Perhaps the authors meant the products of photosynthesis?

Line 346-347: “Thus, the seed setting rate and 1000-grain weight were seriously reduced (Huang et al, 2013).” This phrase should be completed indicating in what conditions the mentioned parameters were seriously reduced.

For the Agronomy journal, references in the text should be given in format [1]. However in this manuscript, references are given in the incorrect format. The “Hanashiro et al” (Line 43), “(Patindoletal,2015)” (Line 242), are absent in the List of references. “Reddy believes…” (Line 64), “(Baoetal.,2010)” (Line 356), “Sui Jiongming et al.” – also needs to be clarified.

Comments on the Quality of English Language

Minor editing of English language required

Author Response

(The authors gave the same response as above.)

Reviewer 3 Report

Comments and Suggestions for Authors

Materials and Methods: provides a detailed description of the experimental setup, procedures, and techniques used in the study. Overall, the section is comprehensive and well-structured, but there are a few areas that could be improved for clarity and completeness.

Proofread the text for grammatical errors and typographical mistakes ("Firgure 1"; "Firgure 3"; "supplementary Firgure S1").

Additional information about the specific characteristics of each region that might influence rice growth and quality would enhance the description of experimental locations and materials.

Including environmental conditions during the growth period, such as temperature and lighting duration, would improve the understanding of the experiment.

While the mention of statistical analysis tools used is appropriate, briefly describing the specific types of statistical tests conducted and their relevance to the study would be helpful.

Results: is detailed and provides valuable insights into the experimental outcomes. However, there are some areas that could be improved for clarity and precision:

Proofread the text for grammatical errors and typographical mistakes ("Previous studies have shown that hardness is positively correlated with AC, while viscosity is negatively correlated with AC (Patindoletal,2015)" could be clarified for better readability).

The analysis of RVA characteristic values provides important insights into the cooking taste quality of rice, but there are missing explanations of abbreviations in Table 2, and the discussion of results could be clearer and more concise.

Discussion: provides valuable insights into the experimental results, but some revisions are needed.

4.1. Effects of Differences in Temperature and Light Conditions: The discussion effectively highlights the average maximum temperature reaching 40.3°C in Anhui province, which was 10-15°C higher than Liaoning and Tianjin. However, it lacks further explanation on how these temperatures affected seed development and the grain-filling process. Additional clarification is needed on how the temperature reduction led to the decrease in grain weight, as stated in the results.

4.2. Relationship between Gelatinization Characteristics and Cooking Taste Quality: More explanation is needed about how RVA values are related to cooking quality, and additional references would strengthen certain claims.

4.3. Effects of Synergistic Effects of Fine Structure and Physicochemical Properties of O Starch: There are inconsistencies that need to be addressed, particularly regarding the "inappropriate molecular weight distribution" for DGR1 in Tianjin and Liaoning.

The conclusion provides a summary of the main findings of the study, but some sentences need refinement to be more precise and clear. Avoid general statements like "balanced conditions" and instead offer specific conclusions based on the research findings.

Comments on the Quality of English Language

Proofread the text for grammatical errors and typographical mistakes 

Author Response

Thank you very much for your comments and suggestions on the article! I will reply to your content in the form of word file
